# Tribological Performance and Thermal Stability of Nanorubber-Modified Polybenzoxazine Composites for Non-Asbestos Friction Materials

**DOI:** 10.3390/polym13152435

**Published:** 2021-07-23

**Authors:** Chanchira Jubsilp, Jakkrit Jantaramaha, Phattarin Mora, Sarawut Rimdusit

**Affiliations:** 1Department of Chemical Engineering, Srinakharinwirot University, Nakhonnayok 26120, Thailand; chanchira@g.swu.ac.th; 2Research Unit in Polymeric Materials for Medical Practice Devices, Department of Chemical Engineering, Faculty of Engineering, Chulalongkorn University, Bangkok 10330, Thailand; sprintfirer@gmail.com (J.J.); phattarin.m@gmail.com (P.M.)

**Keywords:** polymers and plastics, organic matrix composites, thermal analysis, adhesion, brake application

## Abstract

Asbestos-free friction composite based on ultrafine full-vulcanized acrylonitrile butadiene rubber particles (UFNBRPs)-modified polybenzoxazine was successfully developed. The UFNBRPs-modified polybenzoxazine friction composite was characterized for chemical, tribological, and mechanical properties as well as thermal stability. The UFNBRPs not only act as a filler to reduce noise in the friction composites due to their suitable viscoelastic behaviors but also play a key role in friction modifiers to enhance friction coefficient and wear resistance in the polybenzoxazine composites. The chemical bonding formation between UFNBRPs and polybenzoxazine can significantly improve friction, mechanical, and thermal properties of the friction composite. The outstanding tribological performance of the friction composite under 100–350 °C, i.e., friction coefficients and wear rates in a range of 0.36–0.43 and 0.13 × 10^−4^–0.29 × 10^−4^ mm^3^/Nm, respectively, was achieved. The high flexural strength and modulus of the friction composite, i.e., 61 MPa and 6.4 GPa, respectively, were obtained. The friction composite also showed high thermal stability, such as 410 °C for degradation temperature and 215 °C for glass transition temperature. The results indicated that the obtained UFNBRPs-modified polybenzoxazine friction composite meets the industrial standard of brake linings and pads for automobiles; therefore, the UFNBRPs-modified polybenzoxazine friction composite can effectively be used as a replacement for asbestos-based friction materials.

## 1. Introduction

Polymer composite materials composed of different reinforcements and fillers into several engineering polymers to replace metallic and ceramics are currently used in structural applications such as automotive, construction, spacecraft, marine, etc. [1,2,3]. In automotive applications, friction polymer composites such as brake pads are a key brake part because they are the component that contacts and applies pressure and friction to a vehicle’s brake rotors, resulting in stopping the wheel; in consequence, the vehicles stop moving. In the past years, asbestos as a reinforcement for brake pads has been the most widely used in all kinds of vehicles due to its excellent thermal stability, tribological properties, and low cost. However, as a reason of eco-friendly environment, asbestos-free brake pads from polymer composite materials have been developed to replace asbestos ones. Polymer friction materials currently used as brake pads can contain more than 10 different ingredients, which are often categorized into four classes of ingredients, including binder, i.e., phenolic and epoxy resins (10–30 wt%), reinforcing fibers, i.e., glass fiber, aramid fiber, and metallic fiber (5–40 wt%), friction modifiers, i.e., graphite (synthetic and natural graphite), alumina powder and metal oxides (5–35 wt%), and fillers, i.e., barium sulfate, calcium carbonate, cashew dust, and rubber dust (40–80 wt%) [3,4].

Phenolic resins are traditionally used as a binder for friction polymer materials such as brake pads due to their low cost and a suitable combination of mechanical properties and suitable wetting capability with most ingredients [4,5,6,7,8,9]. However, friction polymer materials based on traditional phenolic resin still showed a molding failure, such as cracks generated during a polymerization step because of a gas by-product formed. Moreover, there is a concern about environmental pollution by ammonia as a main component of the gas. Therefore, the use of a novel type of phenolic resin, namely benzoxazine resin, as a binder for friction materials was recently suggested to replace traditional phenolic resin [5,10,11,12,13,14,15,16]. The benzoxazine resin can undergo ring-opening polymerization without catalysts or curing agents and does not release by-products upon curing. In addition, polybenzoxazine has excellent properties such as high thermal stability, flame retardant, high mechanical integrity, and low water absorption [17,18]. However, the disadvantage of polybenzoxazines is their brittleness, which is a typical problem for thermoset resins. Thus, aiming at performance enhancement of polybenzoxazines, studies on copolymers [11,19,20] and fiber/filler-reinforced composites have been extensively performed [21,22,23]. This approach has successfully afforded polybenzoxazines, especially rubber-modified polybenzoxazine, having improved toughness and flexural strength as well as high glass transition temperature in comparison with typical polybenzoxazines. In addition, nitrile butadiene rubber (NBR) in the form of vulcanizates, a vulcanized rubber, is well known for its high friction coefficient and suitable abrasion resistance [24,25,26] as it has been reported that friction material based on ternary binder system of polybenzoxazine, phenolic, and nitrile butadiene rubber (NBR) showed effective ability to stabilize the friction coefficient and wear rate under relatively higher braking temperature due to relatively higher glass-rubber transition temperature compared with polybenzoxazine and phenolic without adding of NBR [16]. However, there is very little or no information about the effect of solid rubber particle contents on tribological behaviors of solid rubber particles-modified polybenzoxazine for friction composites. Therefore, in this work, the outstanding tribological, mechanical, and thermal properties of ultrafine full-vulcanized acrylonitrile butadiene rubber particles (UFNBRPs)-modified polybenzoxazine composites at various UFNBRPs contents were presented. The UFNBRPs-modified polybenzoxazine binder to maintain the friction composite structural integrity under mechanical and thermal stresses was investigated. Moreover, the chemical property and thermal stability were also characterized for understanding the effect of rubber particles on those properties of the friction composite for potential use as brake pads application.

## 2. Materials and Methods

### 2.1. Materials

Benzoxazine resin (BA-a) was based on bisphenol A supported by PTT Phenol Co., Ltd. (Rayong, Thailand), formaldehyde purchased from Merck Co., Ltd. (Darmstadt, Germany), and aniline obtained from Panreac Quimica S.A. (Barcelona, Spain). Aramid short fiber was provided by DuPont (Tokyo, Japan). All chemicals were used as received.

### 2.2. Sample Preparations

Bisphenol A/aniline-based benzoxazine resin (BA-a) was synthesized from bisphenol A, formaldehyde, and aniline according to a patented solventless method [10]. The obtained benzoxazine resin is clear-yellowish solid at room temperature, and the solid benzoxazine resin was then ground to a fine powder and kept in a refrigerator for future use.

Ultrafine full-vulcanized acrylonitrile butadiene rubber particles (UFNBRPs)-modified polybenzoxazine at UFNBRPs contents in a range of 0–15 wt% were prepared. The molding compounds were mixed in an internal mixer at 110 °C for 30 min and thermally cured at 200 °C for 2 h under a hydraulic pressure of 10 MPa by a compression molder. The samples were then air-cooled to room temperature before characterizations.

UFNBRPs-modified polybenzoxazine friction composite was produced from the ingredients listed in Table 1. The compound was prepared by mechanical mixing at 110 °C for at least 15 min to ensure ingredients were wet-out by benzoxazine resin. The compounds were then cured at 200 °C for 2 h in the hydraulic press using a pressure of 34 MPa. The samples were kept in a room to cool them to ambient temperature before any tests were carried out. In addition, UFNBRPs-modified phenolic friction composite was also prepared to compare properties with the polybenzoxazine one.

### 2.3. Research Methods

Differential scanning calorimetry (DSC model 2910, TA instruments, New Castle, DE, USA) was performed under a heating rate of 10 °C/min from 30 to 300 °C, nitrogen purging at 50 mL/min. A small amount of samples ranging from 5 to 10 mg was placed on the aluminum pan with lids, and three samples of each compound were evaluated.

FTIR measurement was conducted at room temperature on a Spectrum GX FTIR spectrometer from Perkin Elmer. The samples were sufficiently thin to be within a range where the Beere–Lambert law is obeyed. All spectra were taken with 64 scans at a resolution of 4 cm^−1^ and a spectral range of 4000–400 cm^−1^. Three samples of each poly(BA-a), UFNBRPs, and poly(BA-a)/UFNBRPs composite were measured.

Thermogravimetric analysis (TGA) from Perkin Elmer (Diamond TG/DTA) was performed under nitrogen purging with a constant flow of 100 mL/min. A sample mass of 10 mg was heated at a linear heating rate of 20 °C/min from room temperature to 900 °C. Three samples of each composite were evaluated.

Dynamic mechanical analyzer (DMA) from Netzsch Inc., model DMA 242 C in the three-point bending mode and a support span of 40 mm was used to examine loss tangent (tanΔ) of the samples. Three samples of each composite with dimensions of 10 × 50 × 3 mm were tested. The tests were performed in a temperature sweep mode with a fixed frequency of 1 Hz. Each sample was tested using a heating rate of 2 °C/min from 30 to 250 °C.

Flexural properties were conducted by a Universal Testing Machine, Model Instron 5567. The measurement was carried out in a three-point bending mode, with a support span of 48 mm and at a crosshead speed of 1.2 mm/min. Five samples with a dimension of 25 × 60 × 3 mm were tested, and the average values for the samples were determined.

A tribological test under dry sliding at a temperature of 25 °C was conducted on a pin-on-disk tribometer having a maximum temperature of 950 °C from CSM Instrument Ltd., Switzerland. The test condition is 10 N for normal load and 0.366 m/s for sliding speed for 1000 m distance. Three samples of each composite with a dimension of 25 × 25 × 7 mm were tested, and the average values for the samples were determined. At a temperature range of 100–350 °C, the sliding speed of the frictional surface of the disk against the test sample shall be a constant speed given within 6 to 8 m/s, and the sliding test under a pressure of 1 MPa according to TIS97-2557 and JIS D 4411 standards. Two samples of each friction composite with a dimension of 25 × 25 × 7 mm were tested, and the average values for the samples were reported.

The worn surface of the samples was evaluated with a JSM5410LV scanning electron microscopy (SEM) from JEOL Ltd. using an acceleration voltage of 15 kV. All samples were coated with a thin film of gold to make the surfaces conductive.

## 3. Results and Discussion

### 3.1. Effect of UFNBRPs Content on Processing Behaviors of BA-a/UFNBRPs Molding Compounds

Curing behavior of benzoxazine resin (BA-a) filled with 0, 2, 5, 10, and 15 wt% of ultrafine full-vulcanized acrylonitrile butadiene rubber particles (UFNBRPs) was studied to be able to control reactivities and optimize processing times of UFNBRPs-filled polybenzoxazine composites. In Figure 1, an exothermic peak of the BA-a was observed at 220 °C. It was attributed to the curing peak of the ring-opening polymerization of its oxazine-ring. The curing peak showed no significant shifting with an increase in UFNBRP contents. In addition, the area under curing peak indicating the heat of polymerization reaction from monomer to polymer (ΔH) was increased with increasing of UFNBRPs in the BA-a, i.e., from 277 J/g for the BA-a to 288, 294, 291, and 288 J/g for adding of UFNBRPs at 2, 5, 10, and 15 wt%, respectively, as listed in Table 2. This expected that phenomenon is related to the chemical reaction between UFNBRPs and poly(BA-a).

In addition, ΔH of BA-a part in each of BA-a/UFNBRP molding compounds is also calculated and listed in Table 1. The ΔH of the BA-a part in each of BA-a/UFNBRP molding compounds was from a multiple of ΔH of the BA-a without UFNBRPs (277 J/g) by weight percentage of the BA-a in the compound. Then, the difference between ΔH of the BA-a without UFNBRPs and ΔH of the BA-a part in each of BA-a/UFNBRP molding compounds is ΔH of UFNBRPs reacted with BA-a. It was found that the ΔH of UFNBRPs reacted with BA-a was increased with an increase in UFNBRP content. This characteristic can confirm that a chemical reaction between UFNBRP and poly(BA-a) was formed.

### 3.2. Chemical Bonding Formation between UFNBRPs and Poly(BA-a)

To further confirm the chemical reaction between UFNBRPs and poly(BA-a), the IR spectra of poly(BA-a), UFNBRPs, and poly(BA-a)/UFNBRPs composite are shown in Figure 2. In Figure 2a, the IR spectrum of the poly(BA-a) presented the absorption bands of a ring-opening of the benzoxazine resin (BA-a) to the poly(BA-a) at 1488 and 878 cm^−1^, and 3320 cm^−1^ due to a tetra-substituted benzene ring mode and the phenolic hydroxyl (–OH) group formation, respectively [11,27,28]. For the UFNBRPs, the spectrum in Figure 2b exhibited the −C≡N stretching at 2233 cm^−1^, while the bands at 2848 and 2925 cm^−1^ were attributed to the C-H stretching vibration of CH_2_ groups in the rubber backbone. The characteristic bands at 969 and 892 cm^−1^, which are the trans-1,4 and 1,2-vinyl characteristic frequencies of the butadiene moieties due to out-of-plane deformation of the =C−H, were also observed. Moreover, the absorption band with a frequency of 1630 cm^−1^ and 1449 cm^−1^ indicated CH_2_ bending vibration of a methylene group, and >C=C< unsaturated sites of the UFNBRPs, respectively [29,30]. In terms of the poly(BA-a)/UFNBRPs composite (Figure 2c), the intensity of the broadband centered at 3320 cm^−1^ due to the –OH group tended to decrease when compared with that of the poly(BA-a), implying to the consumption of the –OH group in the poly(BA-a). While the new band at 1107 cm^−1^ assigned to ether linkages (C−O−C stretching) formed between the –OH group in the poly(BA-a) and the −CH=CH_2_ groups of the UFNBRPs [11] was observed. In addition, the band at 1647 cm^−1^ typical of oscillations of the C=N group also appears in the spectrum, while the intensity of the band at 1630 and 2233 cm^–1^ typical of oscillations of the >C=C< and C≡N groups, respectively, disappeared. Therefore, all appearance bands of the poly(BA-a) filled with UFNBRPs are clear evidence of the chemical bonding formed between the poly(BA-a) and the UFNBRPs.

### 3.3. Thermomechanical Property, Thermal Stability, and Flammability Poly(BA-a)/UFNBRPs Composites

Thermomechanical property, i.e., storage modulus at 30 °C (E′) and glass transition temperature (T_g_) calculated by loss modulus of the poly(BA-a)/UFNBRPs composites at various UFNBRPs contents are summarized in Table 3. As expected, the T_g_s of the composites was enhanced by the addition of UFNBRPs due to chemical bonding formation between poly(BA-a) and UFNBRPs as discussed in FTIR analysis, resulting in crosslink density improvement. The increased T_g_ with the addition of UFNBRPs of poly(BA-a)/UFNRPs composites would lead to substantial enhancement of friction force and specific wear of poly(BA-a)/UFNBRPs friction composite since the relatively higher T_g_ (or higher glass-rubber transition temperature), resulting in better ability to stabilize the tribological property, i.e., friction coefficient and wear rate, under relatively higher braking temperature [16]. It was observed that the higher content of poly(BA-a) of the composites showed a higher storage modulus. This behavior was well known that the modulus of the poly(BA-a), a solid material was higher than that of UFNBRPs, which is a softened material at the same temperature. Therefore, it was expected that the higher poly(BA-a) content of the poly(BA-a)/UFNBRPs composite acted as a matrix of the poly(BA-a)/UFNBRPs friction composite can help improve the dimension stability of the friction composite.

From Table 3, it was also found that the degradation temperature (T_d5_ and T_d10_) and char yield of the poly(BA-a)/UFNBRPs composites tended to be quite stable with an addition of UFNBRPs up to 5 wt% followed by a sharp decrease with the incorporation of UFNBRP more than 5 wt% UFNBRPs. The obtained degradation temperature and char yield of the composites, i.e., poly(BA-a)/UFNBRPs (2 wt%) and poly(BA-a)/UFNBRPs (5 wt%) was still similar, indicating that the composites should have a high degree of crosslinking, which requires more energy for degradation. Moreover, the formation of char is an effective method of increasing the fire resistance of polymers as it can also be confirmed by the limiting oxygen index (LOI), representing the minimum concentration of oxygen, expressed as a percentage, that will support ignition of a polymer. LOI was determined with Van-Krevelen and Hoftyzer equation [31] when CR is char yield value.
LOI = 17.5 + 0.4 CR (1)

LOI values of the poly(BA-a)/UFNBRPs composites at various UFNBRPs contents were in the range 24.9–27.6, as the results are summarized in Table 2. The obtained LOI values indicated that the addition of UFNBRPs up to 5 wt% has self-extinguishing properties since the composites showed LOI above the threshold value of 26.

### 3.4. Tribological Properties of UFNBRPs-Modified Poly(BA-a) Composite at 25 °C

Average friction coefficient (μ_avg._) and wear rate at 25 °C under dry sliding friction using pin-on-disk of poly(BA-a)/UFNBRPs composites at various UFNBRPs were measured and depicted in Figure 3. It was observed that the μ_avg._ of the composites was sharply increased from 0.50 to 0.60 in a range of UFNBRP contents from 0 to 5 wt%. This behavior can be suggested that the development of friction force at the sliding surface for the poly(BA-a) and the poly(BA-a)/UFNBRPs composites pursues different mechanisms. UFNBRPs can be deformed on the composite’s contact surface to generate friction force by an exchange with thermal energy after applied load and velocity. As a consequence, the μ_avg._ increased with an increase in the actual contact area and the surface roughness. The addition of greater than 5 wt% UFNBRPs revealed a slight increase in the μ_avg._ of the composites. It is possible that an agglomerate formation of UFNBRPs changed in the surface roughness, which resulted in a significant reduction in the actual contact area between rigid asperity and the composite surface. Furthermore, it was also found that the addition of UFNBRPs up to 5 wt% significantly improved the wear resistance of the poly(BA-a) as the wear rate was decreased from 3.35 × 10^−4^ mm^3^/Nm for poly(BA-a) to 1.83 × 10^−4^ mm^3^/Nm for poly(BA-a)/UFNBRPs (5 wt%) composite. The addition of UFNBRPs greater than 5 wt% resulted in an increase in the wear rate of the composites. It seems like that the 5 wt% UFNBRPs, having uniform dispersion in the poly(BA-a) is optimum content for the poly(BA-a) to be a binder for poly(BA-a)/UFNBRPs friction composite.

### 3.5. Tribological Property of UFNBRPs-Modified Poly(BA-a) Composite at High Temperatures

The influence of temperature should be paid special attention to since the different temperatures will cause a great change in the friction coefficient. To investigate the effect of temperature on the tribological property of the poly(BA-a) modified with UFNBRPs, the poly(BA-a)/UFNBRPs (5 wt%) composite was selected to measure μ_avg._ and wear rate under dry sliding friction using pin-on-disk at a temperature range of 100 to 350 °C and listed in Table 4. The μ_avg._ showed a slight decrease from 0.60 to 0.50 in a temperature range of 100 to 250 °C. It is possible that the heat generated due to the tribological test increased the interface temperature and this, in turn, decreased the μ_avg._. In addition, the decrease in the μ_avg._ may be caused by a greater degree of thermal softening and plastic flow of the UFNBRPs at higher temperatures and the change state effect from glass state to rubber state at T_g_ (182 °C) of the poly(BA-a)/UFNBRPs (5 wt%) composite, which may have an effect on adhesion ability of the poly(BA-a) and UFNBRPs.

In terms of the wear rate of the composite, it can be seen that the wear rate was very different, and they generally became more pronounced with temperature. The wear rate slowly increased with the temperature up to 200 °C, and then rapidly increased as the temperatures increased above 200 °C. This characteristic was due to the lower thermal stability of the composite at a temperature of above 200 °C. Moreover, it is possible that a portion of friction energy was converted to heat, and the remainder resulted in plastic deformation, micro-cracks, and a change in surface roughness.

Scanning electron microscope (SEM) photographs of the worn surface of poly(BA-a)/UFNBRPs (5 wt%) composite at different temperatures are illustrated in Figure 4. The wear mechanisms at each temperature were discussed by analyzing the morphologies of their worn surfaces. In Figure 4a, the worn surface of the composite at 100 °C showed a suitable dispersion of UFNBRPs in the poly(BA-a) matrix and regions of uniform roughness. The deformation of the UFNBRPs, micro-plowing, and loose fine wear debris particles were observed. These suggested that the wear process was governed by the material deformation and mechanical micro-plowing type of abrasive wear, leading to obtaining high friction coefficient of the composite as a higher friction force was developed at the interface. At 150 °C, further changes in the wear mechanism occur, as can be seen in Figure 4b. The brittle fracture is not seen along the sides of the wear tracks, and the friction film was formed on the worn surface; in consequence, the dominant wear mechanism changed to the adhesion. This phenomenon resulted in a greater lubricity effect of contact surfaces, consequently lower the frictional response. The surface delamination due to plastic deformation was also observed. These characteristics cause an increase in the wear rate of the composite. Figure 4c revealed no wear debris on the worn surface, while the surface delamination formation at 200 °C was observed. The scale-like damage pattern on the surface generated under a repeating load during sliding was also observed as a responsible variable for the higher wear rate of the composite. These characteristics suggested that the wear process was governed mainly by an adhesive wear mechanism. At 250 °C, in Figure 4d, the damage of the worn surface is more severe than that observed at 200 °C. In addition, the conventional-sized wear debris formation due to deformation and cracking was formed on the worn surface and contributed to a rapid increase in the wear rate of the composite by increasing temperature. It is well known that abrasion and adhesion wear mechanisms are required and the most important parameters in the braking of automobiles [32]. Therefore, it is possible that the obtained poly(BA-a)/UFNBRPs composite is attractive for brake pads application with required wear mechanisms at the desired temperature.

### 3.6. Properties of Poly(BA-a)/UFNBRPs Composite for Non-Asbestos Friction Material

As all results above mentioned was analyzed, it was observed that the amount of UFNBRPs content for poly(BA-a)/UFNBRPs composite should not be more than 5 wt%, showing the balanced properties between tribological and thermal properties, and also base on the rubber content in friction composites designed by Akebono Brake Industry Co Ltd. for friction composite materials [5]. As a consequence, the poly(BA-a)/UFNBRPs friction composite was prepared using the formulation listed in Table 1 to investigate the properties of a friction material based on poly(BA-a) and UFNBRPs. In comparison, the phenolic/UFNBRPs friction composite was also prepared and measured properties. μ_avg._ and wear rate values of both friction polymer composites at various temperatures are plotted in Figure 5. The μ_avg._, i.e., 0.40–0.43 of poly(BA-a)/UFNBRPs friction composite as the temperature increased from 100 to 300 °C was obtained, and the μ_avg._ of the friction composite was then reduced to 0.36 at 350 °C. It showed the same behavior observed for poly(BA-a)/UFNBRPs (5 wt%) composite that the μ_avg_ was clearly decreased at elevated temperature. It is possible that the temperature of 350 °C is higher than the T_d5_ of the poly(BA-a), i.e., 325 °C; in consequence, the wear debris was formed, resulting in a decrement of friction coefficient of the poly(BA-a)/UFNBRPs friction composite. However, the obtained μ_avg._ values of the friction composite still meet the required standard for brake pads, i.e., μ_avg._ at 100–350° is in a range of 0.2–0.7 [33,34]. In addition, the μ_avg._ at the various temperatures of the poly(BA-a)/UFNBRPs friction composite showed higher than that of the phenolic one, which was in a range of 0.36–0.41. This characteristic suggested that the adhesion between the poly(BA-a)/UFNBRPs friction composite and the brake disc showed more effectiveness than that between the phenolic type and the brake disc. Moreover, in comparison, the friction coefficient of the poly(BA-a)/UFNBRPs friction composite with that of commercial asbestos-based friction material. It was found that the poly(BA-a)/UFNBRPs friction composite showed a more stable friction coefficient against the different temperatures than the asbestos-based friction because of the inherent thermal stability of the poly(BA-a)/UFNBRPs matrix.

The wear performance of the poly(BA-a)/UFNBRPs friction composite was measured in terms of wear rate and also depicted in Figure 5. The wear rate at different temperatures of the poly(BA-a)/UFNBRPs friction composite was also agreed with the values according to the standard of brake lining for vehicles, i.e., wear rate is less than 0.5 × 10^−4^ mm^3^/Nm at 100 °C and less than 3.5 × 10^−4^ mm^3^/Nm at 350° [34,35]. Moreover, the poly(BA-a)/UFNBRPs friction composite showed higher wear resistance in the temperature range of 100–350 °C compared to the phenolic and asbestos ones, respectively. It is possible that outstanding wear resistance of the poly(BA-a)/UFNBRPs friction composite has been achieved by the suitable interfacial adhesion between the UFNBRPs-modified poly(BA-a) and the ingredients. As a consequence, the development of friction force at the sliding surface for the poly(BA-a)/UFNBRPs friction composite may be a different wear mechanism compared with the phenolic and asbestos ones.

Mechanical properties, i.e., flexural strength (σ_f_) and flexural modulus (E_f_), are important for friction polymer composites used for brake friction materials. From Table 5, it was found that the flexural strength and flexural modulus of the poly(BA-a)/UFNBRPs friction composite was about 61 ± 4 MPa and 6.4 ± 0.3 GPa, respectively, which are in agreement with previous publications [35]. It has been reported in the range of 10–55 MPa for flexural strength and 3.0–8.0 GPa for flexural modulus. Moreover, as expected, the flexural strength and flexural modulus of the poly(BA-a)/UFNBRPs friction composite was higher than those of the phenolic one, i.e., σ_f_ = 33 ± 2 MPa and E_f_ = 3.3 ± 0.2 GPa. This characteristic can be confirmed that the UFNBRPs-modified poly(BA-a) has higher interfacial adhesion to the ingredients than that of the phenolic one. In addition, it is possible that the flexural strength of the poly(BA-a), i.e., 126 MPa [18], was higher than that of the phenolic for general-purpose type, i.e., 28–75 MPa [36,37].

In general, the ideal brake friction material should have a constant friction coefficient under various operating conditions such as temperature and applied loads. In this work, the thermal properties such as T_d5_ and T_g_ measured from loss tangent of dynamic mechanical analysis of the UFNBRPs-modified poly(BA-a) friction composite was investigated, as also listed in Table 5. The T_d5_ and T_g_, i.e., 410 and 215 °C, respectively, of the poly(BA-a)/UFNBRPs friction composite was found to be similar to those of the phenolic one, i.e., T_d5_ = 424 °C and T_g_ = 210 °C. The values can be related to the thermal degradation of the organic binder resin of the friction composite. It can be implied that the friction composites may show substantial brake fade resistance. It describes the status of friction stability under severe temperature conditions, and it refers to the loss of brake effectiveness due to excessive frictional heat from the extended brake applications [38]. In addition, the high-temperature wear of organic friction materials was also found to be related to the thermal degradation of the organic ingredients. It is possible that the high-temperature wear of both friction materials was controlled by thermal degradation of organic binder resin as it is for organic friction materials.

## 4. Conclusions

The evaluation of friction polymer composites based on ultrafine full-vulcanized acrylonitrile butadiene rubber particles (UFNBRPs)-modified polybenzoxazine can be summarized as follows. The UFNBRPs-modified polybenzoxazine friction composites showed an outstanding friction coefficient and wear rate. The significant mechanical properties, i.e., flexural strength and modulus, were obtained due to suitable interfacial adhesion between the UFNBRPs-modified polybenzoxazine and all ingredients. The high thermal stability, resulting in a better ability to stabilize the friction coefficient and wear rate under relatively higher temperatures, can help better design friction composites. In addition, the ether bonds formed between the polybenzoxazine and the UFNBRPs are attributed to the significant improvement in tribological, mechanical, and thermal properties of the obtained UFNBRPs-modified polybenzoxazine friction composites.

## Figures and Tables

**Figure 1 polymers-13-02435-f001:**
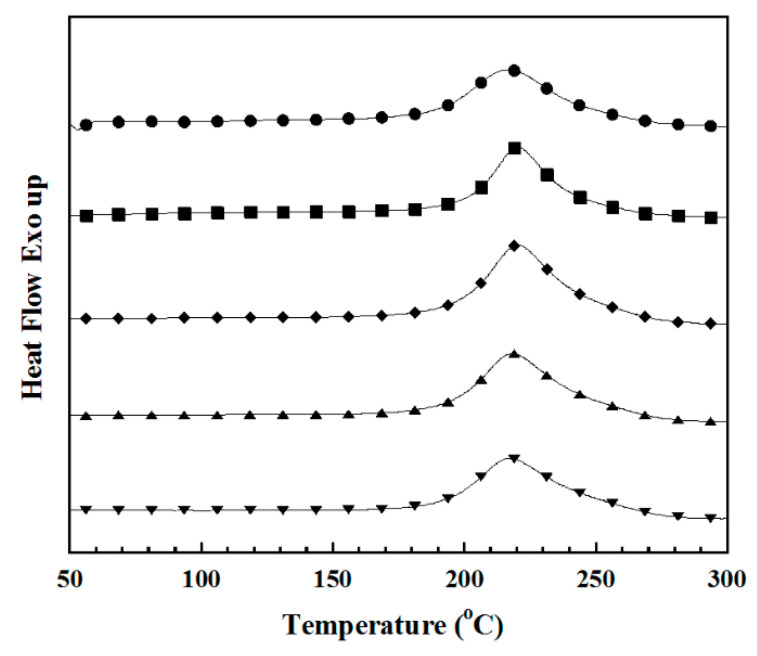
DSC thermograms of BA-a/UFNBRP molding compounds at different UFNBRPs contents: (●) 0 wt%, (■) 2 wt%, (◆) 5 wt%, (▲) 10 wt%, (▼) 15 wt%.

**Figure 2 polymers-13-02435-f002:**
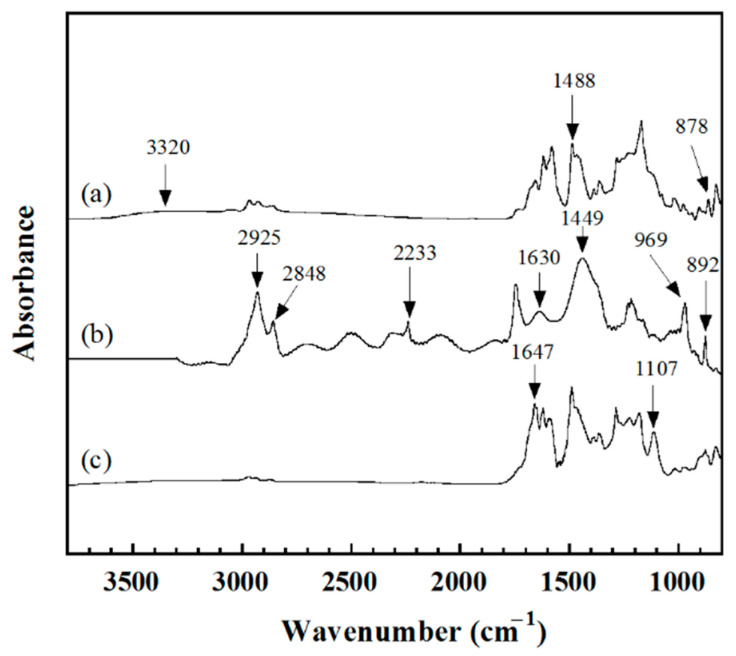
FTIR of (a) poly(BA-a), (b) UFNBRPs, (c) poly(BA-a)/UFNBRPs (15 wt%) composite.

**Figure 3 polymers-13-02435-f003:**
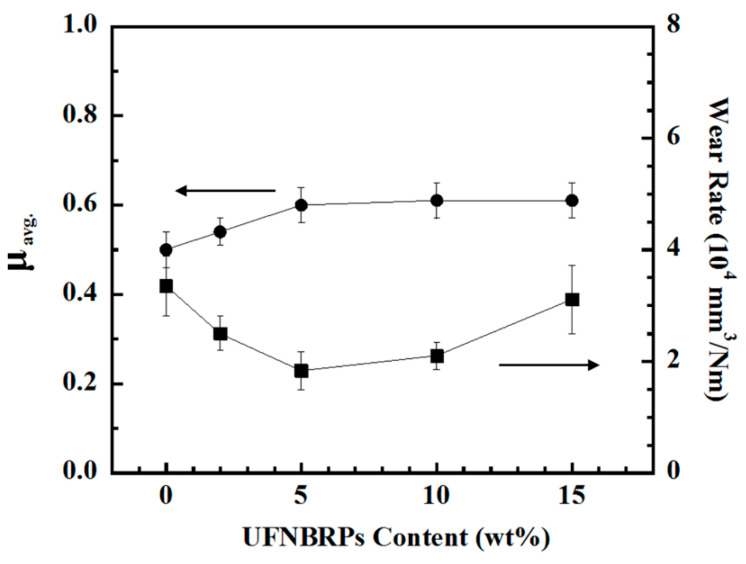
Tribological properties of poly(BA-a)/UFNBRPs composites at various UFNBRPs contents at 25 °C: (●) μ_avg._, (■) wear rate.

**Figure 4 polymers-13-02435-f004:**
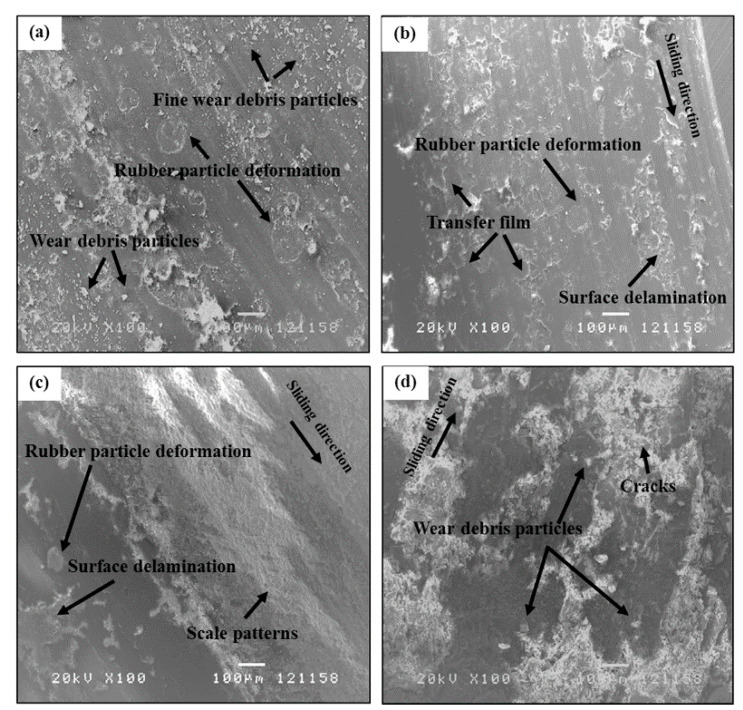
SEM micrographs of worn surface of poly(BA-a)/UFNBRPs (5 wt%) composite at different temperatures: (**a**) 100 °C, (**b**) 150 °C, (**c**) 200 °C, (**d**) 250 °C.

**Figure 5 polymers-13-02435-f005:**
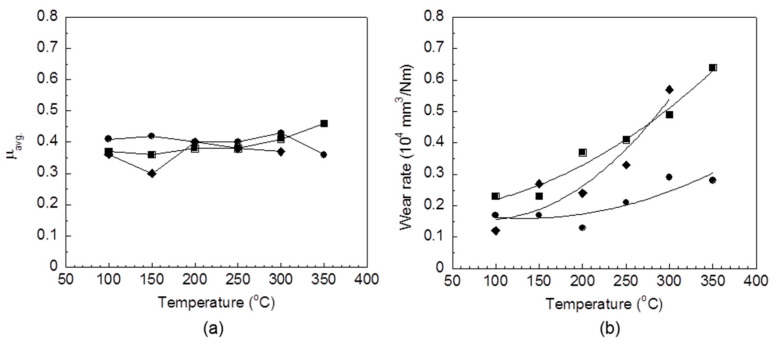
(**a**) μ_avg._ and (**b**) wear rate of polymer friction composites: (●) poly(BA-a)/UFNBRPs friction composite, (■) phenolic/UFNBRPs friction composite, (◆) commercial brake pad (asbestos-based).

**Table 1 polymers-13-02435-t001:** Formulation of friction polymer composites used in this investigation.

Ingredients (wt%)	Poly(BA-a)/UFNBRPs Friction Composite	Phenolic/UFNBRPs Friction Composite
Benzoxazine resin	10	-
Phenolic resin with curing agent	-	10
Aramid fiber	5	5
Inorganic fiber	35	35
Graphite	10	10
Zirconium silicate	7	7
Barium sulfate	25	25
UFNBRPs	4	4
Cashew dust	4	4

**Table 2 polymers-13-02435-t002:** Approximated heat of polymerization reaction of BA-a/UFNBRPs molding compounds.

UFNBRP Content (wt%)	ΔH of BA-a/UFNBPs Compound (J/g)	ΔH of BA-a in Compound (J/g)	ΔH of Reacted UFNBRPs in Compound (J/g)
0	277	277	0
2	285	271	14
5	291	263	28
10	288	249	39
15	290	235	55

**Table 3 polymers-13-02435-t003:** Thermal stability, thermomechanical property, and flammability poly(BA-a)/UFNBRPs composites at various UFNBRPs contents.

UFNBRP Content (wt%)	T_g_ (°C)	E′ at 30 °C (GPa)	T_d5_ (°C)	T_d10_ (°C)	Char yield at 800 °C (%)	LOI
0	170	5.2	325	345	25.3	27.6
2	175	4.7	323	345	23.6	26.9
5	182	3.7	321	346	23.6	26.9
10	185	3.3	317	342	20.9	25.9
15	188	2.8	304	340	18.4	24.9

**Table 4 polymers-13-02435-t004:** Friction coefficient and wear rate of poly(BA-a)/UFNBRPs (5 wt%) composite at various temperatures.

	100 °C	150 °C	200 °C	250 °C	300 °C *	350 °C *
μ_avg._	0.61 ± 0.02	0.58 ± 0.03	0.55 ± 0.01	0.50 ± 0.01	na	na
Wear rate (mm^3^/Nm)	3.98 × 10^−4^	4.23 × 10^−4^	11.13 × 10^−4^	28.81 × 10^−4^	na	na

* The thickness of the composite sample was not in the required standard for testing.

**Table 5 polymers-13-02435-t005:** Mechanical and thermal properties of polymer friction composites.

Samples	E_f_ (GPa)	σ_f_ (MPa)	T_g_ (°C)	T_d5_ (°C)	Char Yield at 800 °C (%)
Poly(BA-a)/UFNBRPs-based composite	6.4 ± 0.3	61 ± 4	215	410	80
Phenolic/UFNBRPs-based composite	3.3 ± 0.2	33 ± 0.2	210	424	79

## Data Availability

Data is contained within the article.

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
