# Peer review of "Tribological Performance and Thermal Stability of Nanorubber-Modified Polybenzoxazine Composites for Non-Asbestos Friction Materials"

_polymers, 2021, doi:10.3390/polym13152435_

Round 1

Reviewer 1 Report

The manuscript describes the production of novel brake pads aiming at substituting asbestos fibers with nitrile butadiene rubber (NBR) nanoparticles resulting in a more ecofriendly material. Chemical and mechanical properties of the material and possible application in automotive industry were evaluated. Discussion of some of the results should be improved as indicated.

1) Polymer thermal stability. As written at the beginning of section 3.1, authors are investigating the change in the chemical and thermal properties of benzoxazine resin induced by the addition of ultra-fine full-vulcanized acrylonitrile butadiene rubber particles (UFNBRPs). Table 3 summarizes the properties of the material at increasing content of the UFNBRPs. As it appears, Td, Char yield, E’ and Tg decrease with increasing the UFNBRPs content leading to a degradation of the material properties while only LOI improve. This does not appear clearly in the description given by authors. In addition, it is not clear why authors refer to a 5% weight loss and what this percentage represents.

2) It would be interesting to show the TGA spectra of the phenolic resin, and the modified phenolic-polyphenolic UFNBRPs modified materials.

3) Table 4: authors display the friction and wear rate of the new brake pads at different temperatures. It would help to report the values of conventional pads to better compare the performances of the new materials with the conventional ones

4) SEM analysis of the worn surfaces. Following the results summarized in table 4, the SEM images show that increasing the temperature the surface degradation increases. The conclusion drawn by authors is “substantial durability of the composite at the elevated temperature”. Authors deduce a good resistance of the material because presence of the resin but this conclusion is not supported by the increased wear coefficient at the same temperature. Please better discuss results correlating SEM analysis and values of table 4.

Author Response

Reviewer#1

Comments and Suggestions for Authors

The manuscript describes the production of novel brake pads aiming at substituting asbestos fibers with nitrile butadiene rubber (NBR) nanoparticles resulting in a more ecofriendly material. Chemical and mechanical properties of the material and possible application in automotive industry were evaluated. Discussion of some of the results should be improved as indicated.

1.1) Polymer thermal stability. As written at the beginning of section 3.1, authors are investigating the change in the chemical and thermal properties of benzoxazine resin induced by the addition of ultra-fine full-vulcanized acrylonitrile butadiene rubber particles (UFNBRPs). Table 3 summarizes the properties of the material at increasing content of the UFNBRPs. As it appears, Td, Char yield, E’ and T  decrease with increasing the UFNBRPs content leading to a degradation of the material properties while only LOI improve. This does not appear clearly in the description given by authors.

Authors’ Response: Thank you very much for your valuable comment at this point. The authors agree with the reviewer and the section 3.1 was revised to be clear by focusing on thermal behavior, i.e. glass transition temperature. Since it has been reported that the glass transition temperature (glass-rubber transition temperature) of polymer matrix can influence the tribological property of the friction composites. It is noted that the friction material showed higher glass transition temperature, resulting in better capability to stabilize the friction coefficient under relatively higher applied temperature [Ref.: Wu, Y.; Zeng, M.; Xu, Q.; Hou, S.; Jin, H.; Fan, L. Effects of glass-to rubber transition of thermosetting resin matrix on the friction and wear properties of friction materials. Tribol. Int. 2012, 54, 51–57.]. Moreover, the dimension stability of the friction composite enhanced by an increase of poly(BA-a) content has also been informed. The revision of the section showed in Page 6 (highlighted in yellow).

In case Td of the composites, Td at 10 % weight loss has also been added. It was found that the trend of Td of the composites was still similar by the addition of 2 and 5 wt% UFNBRPs. The revision of thermal stability of the composites can be seen in Page 6 (highlighted in blue).   

1.2) In addition, it is not clear why authors refer to a 5% weight loss and what this percentage represents.

Authors’ Response: Thank you very much for your question.

In general, to compare degradation temperature of the composite/friction composites, their degradation temperature was reported at 1, 5, and 10 % weight loss as can be seen in Cai et al (2015), Fakhraddin (2021), and Lertwassana (2019). If the composite shows higher degradation temperature at the lower percentage of weight loss, it means that the composite shows more stable than the composite having higher degradation temperature at the higher percentage of weight loss.

Cai, P. et al., Effect of resins on thermal, mechanical and tribological properties of friction materials, Tribology International 87 (2015) 1–10

Fakhraddin Y.F., Thermal behavior of N-Methylaniline modified phenolic friction composites, Polymers and Polymer Composites,

https://doi.org/10.1177/09673911211020718

Lertwassana, W. et al., High performance aramid pulp/carbon fiber-reinforced polybenzoxazine composites as friction materials. Compos. Part B 2019, 177, 107280.

2) It would be interesting to show the TGA spectra of the phenolic resin, and the modified phenolic-polyphenolic UFNBRPs modified materials.

Authors’ Response: Thank you very much for your comment.

In this work, TGA thermogram of the phenolic resin and the phenolic modified with UFNBRPs has not been investigated, while TGA data of the friction material based on UFNBRPs-modified phenolic acted as a matrix have been evaluated to compare with our friction material (poly(BA-a)/UFNBRPs based). However, the effect of rubber, i.e. NBR or SBR on thermal degradation of phenolic resin/phenolic composite has been reported by Surojo et al. (2019), Liu et al. (2015), and Cai et al. (2015). The results showed that the addition of rubber particles, i.e. NBR or SBR at low content (approximately 2-8 wt%) can help slightly improve degradation temperature of phenolic resin/phenolic composite due to optimal increased crosslink density.

Surojo, E. et al., Effect of nitrile butadiene rubber (NBR) on mechanical and tribological properties of composite friction brakes, Tribology in Industry, 41 (2019), 516-525.

Liu, W.W. et al., The toughening effect and mechanism of styrene–butadiene rubber

nanoparticles for novolac resin, J. Appl. Polym. Sci., 132 (2015), 41533.

Cai, P. et al., Effect of resins on thermal, mechanical and tribological properties of friction materials, Tribology International, 87 (2015), 1–10.

3) Table 4: authors display the friction and wear rate of the new brake pads at different temperatures. It would help to report the values of conventional pads to better compare the performances of the new materials with the conventional ones

Authors’ Response: Thank you very much for your comment.

Table 4 shows friction coefficient and wear rate at different temperature for poly(BA-a)/UFNBRPs composite acted as a polymer matrix for poly(BA-a)/UFNBRPs friction composite (new brake pads). While, the friction and wear rate of the new brake pads at different temperatures depicted in Figure 5 and compared to the phenolic brake pads prepared by our team. As suggested by reviewer, the friction coefficient and wear rate of conventional one (commercial brake pad, asbestos based) has also been plotted in Figure 5 to compare with the new one. The comparison of friction coefficient and wear rate has been discussed in the section 3.5, Page 9, 10 (highlighted in green)

4) SEM analysis of the worn surfaces. Following the results summarized in table 4, the SEM images show that increasing the temperature the surface degradation increases. The conclusion drawn by authors is “substantial durability of the composite at the elevated temperature”. Authors deduce a good resistance of the material because presence of the resin but this conclusion is not supported by the increased wear coefficient at the same temperature. Please better discuss results correlating SEM analysis and values of table 4.

Authors’ Response: Thank you very much for your comment.

To better discuss results correlating SEM analysis and tribological values in Table 4, the sentence of “However, the resin-rich surface layer was still observed. This suggested substantial durability of the composite at the elevated temperature.” was deleted and the wear behavior on the worn surface of the composite in Figure 4(d) has been modified.

Reviewer 2 Report

The review concerns the work entitled Tribological performance and thermal stability of nanorubber-modified polybenzoxazine composites for non-asbestos friction materials.

My comments are below:

  • Chapter 2.3. The research instruments were described in this chapter. Therefore this chapter should be entitled 'Research methods' or 'Research instruments'
  • Please write how many samples have been studied?
  • How many tribological tests have been performed?
  • Did the Authors take the images of samples before tribological tests?
  • Figure 4. Presented images are of poor quality - descriptions are unreadable.

Author Response

Reviwer#2

1) Chapter 2.3. The research instruments were described in this chapter. Therefore this chapter should be entitled 'Research methods' or 'Research instruments'

Authors’ Response: Thank you very much for your suggestion.

The title of ‘Sample Characterizations’ in section 2.3 (Page 3) has been replaced by ‘Research Methods’ as suggested by reviewer.

2) Please write how many samples have been studied?

Authors’ Response: Thank you very much for your comment.

The number of samples of each research method has been added in section of 2.3 Research Methods, Pages 3, 4 (highlighted in pink).

3) How many tribological tests have been performed?

Authors’ Response: Thank you very much for your comment.

The tribological property at 25oC were evaluated from three samples of each composite and the results has been plotted in Figure 3 (Page 7). While, the experimental of tribological property at high temperature (100-350oC) according to Thailand Industrial Standard (TIS97 2557 (2014): Brake lining for vehicles has been shown in section 2.3 Research Methods. The standard is similar to Japan Industrial Standard (JIS D 4411: Brake Linings and Pads for Automobiles). In general, two samples used to measure tribological properties, i.e. coefficient of friction and wear rate for each time. Therefore, the results reported in Figure 5 was the averaged values. The tribological property at high temperature of composites and friction composites has been listed in Table 4 (Page 8) and depicted in Figure 5 (Page 10), respectively.

4) Did the Authors take the images of samples before tribological tests?

Authors’ Response: Thank you very much for your question.

The image of samples before tribological testes was not taken. However, the surface of the sample before tribological test can be seen in Figure 4b (in red circle). The sample surface before tribological test looks smoother than the worn surface of the sample after the test.

5) Figure 4. Presented images are of poor quality - descriptions are unreadable.

Authors’ Response: Thank you very much for your suggestion.

The resolution of Figure 4 has been improved as suggested by reviewer. 

Round 2

Reviewer 1 Report

//